# Insight on Infections in Diabetic Setting

**DOI:** 10.3390/biomedicines11030971

**Published:** 2023-03-21

**Authors:** Bianca Pari, Matteo Gallucci, Alberto Ghigo, Maria Felice Brizzi

**Affiliations:** Department of Medical Sciences, University of Turin, Corso Dogliotti 14, 10126 Turin, Italy

**Keywords:** Diabetes Mellitus, infection, tuberculosis, anti-diabetic drugs

## Abstract

The correlation between diabetes mellitus and infectious diseases is widely recognized. DM patients are characterized by the impaired function of the immune system. This translates into the occurrence of a variety of infections, including urinary tract, skin and surgical site infections, pneumonia, tuberculosis, and, more recently, SARS-CoV-2. Hyperglycemia has been identified as a relevant factor contributing to unfavorable outcomes in hospitalized patients including SARS-CoV-2 patients. Several studies have been performed proving that to maintain the proper and stringent monitoring of glycemia, a balanced diet and physical activity is mandatory to reduce the risk of infections and their associated complications. This review is focused on the mechanisms accounting for the increased susceptibility of DM patients to infections, with particular attention to the impact of newly introduced hypoglycemic drugs in sepsis management.

## 1. Introduction

Diabetes mellitus (DM) is a chronic disease characterized by abnormal blood glucose levels resulting from impaired insulin action and/or insulin secretion, usually both [1]. DM can be classified as type 1 diabetes, type 2, and gestational diabetes [2]. According to the 10th edition of the IDF Diabetes Atlas, 536.6 million people are currently diagnosed with DM worldwide with a prevalence of 10.5%, and this number is expected to increase to 783.2 million in 2045. The prevalence of DM increases with age, while the incidence of type 2 diabetes is expected to decrease or remain stable in high-income countries [3]. In addition to macro- and microvascular complications, an increased risk of infection is commonly associated with DM [4]. Individuals with DM are at a greater risk of hospitalization and mortality due to viral, bacterial, and fungal infections [5]. However, recent evidence indicates that DM does not represent a significant risk factor for poor survival in patients with sepsis, regardless of intensive care unit (ICU) admission [6,7]. Nevertheless, DM and sepsis remain important causes of morbidity and mortality worldwide, and DM patients represent the largest population experiencing post-sepsis complications [8]. This is mainly due to immunosuppression and uncontrolled hyperglycemia. In fact, high blood glucose impairs innate and adaptive immunity through various mechanisms [9].

Poor glycemic control increases the risk for skin, bone, eye, ear, gastrointestinal, urinary tract, and respiratory infections [10]. Moreover, impaired healing of diabetic wounds, which affects approximately 25% of all DM patients, is associated with an increased risk of limb amputation, thereby representing a crucial economic and psychosocial issue [11]. Uncommon life-threatening infections are also more frequent among DM patients. These include invasive otitis externa, rhino-cerebral mucormycosis, and emphysematous infections of the gall bladder, kidney, and urinary bladder [12]. Evidence has also been provided on the prevalence of drug resistance in DM patients [13], however, an increased prevalence of resistance to commonly used antibiotics in DM patients is still debated [14]. More recently, several infections have been supposed to rely on newly introduced therapies. As an example, sodium-glucose co-transport 2 inhibitors (SGLT2-i) are associated with the occurrence of urinary and genital tract infections [15,16,17]. Moreover, among infections commonly associated with hospitalization, including urinary tract and skin infections, pneumonias, and surgical infections, the presence of DM confers an increased risk (Figure 1).

These patients are more susceptible to urinary tract infections than non-DM individuals. They have a higher risk of developing asymptomatic bacteriuria, cystitis, acute pyelonephritis, and complications such as emphysematous pyelonephritis [9,18]. Bacteria isolated from DM patients with an urinary tract infection (UTI) do not differ from those found in non-DM patients with complicated UTI. *E. coli* are the most common pathogens, while *Klebsiella* spp., *Enterobacter* spp., *Proteus* spp., Group B *Streptococci*, and *Enterococcus faecalis* are the most frequently isolated pathogens [19]. DM patients are more likely to develop skin and soft tissue infections, including cellulitis and osteomyelitis [14]. *Staphylococcus aureus* and *Pseudomonas* [20] are the most common isolated gram-positive and negative bacteria respectively. Methicillin-resistant *Staphylococcus aureus* (MRSA) and other antibiotic-resistant pathogens generally account for skin and soft tissue infections in the diabetic foot compared to other tissue sites and populations [14]. DM increases the susceptibility to different respiratory infections, thereby representing an independent risk factor for lower respiratory tract infections, particularly influenza and pneumonia [20,21]. More importantly, DM individuals are at higher risk of pulmonary infections caused by microorganisms such as *Mycobacterium tuberculosis*, *Staphylococcus aureus*, gram-negative bacteria, and fungi, and have a high risk of hospitalization upon influenza or flu-like infections. Additionally, infections caused by *Streptococcus* pneumonia or influenza virus are characterized by high morbidity and mortality rates [20]. Furthermore, pulmonary infections in elderly DM patients remain occult. Advanced age, comorbidities (senile dementia, hypothyroidism), and prolonged bed rest are indeed considered independent risk factors for occult pneumonia [22], resulting in long-term hospitalization and increased mortality. DM also increases community-acquired pneumonia (CAP) compared to non-diabetic individuals, and gram-negative bacteria such as *K. pneumoniae* and *S. aureus* are much more commonly isolated [9]. The susceptibility to fungal infections caused by Mucorales has been estimated at 75% in this population. Aspergillus is an additional microorganism causing infections in these patients [23]. It has been reported that *S. pneumoniae*, *Enterobacter*, *K. pneumoniae*, *Serratia*, *E. coli*, *S. aureus*, *Proteus,* and *Haemophilus influenzae* are the most common bacteria causing hospital-acquired pneumonia (HAP) during the first four days of hospitalization, while *Acinetobacter*, MRSA, *E. coli*, L. pneumophila, *Pseudomonas aeruginosa*, and *K. pneumonia* are more prevalent after day five [20,24]. Tuberculosis infections are also frequent and increased with a mortality rate corresponding to 50% [25]. Hyperglycemia also predisposes to superinfection of the surgical site following surgery (SSI), and the association between pre- and post-surgery hyperglycemia remains a significant risk factor for SSI [26]. In conclusion, the risk and mortality associated with infectious diseases are high in DM patients, implying that infections should be considered among the most common DM complications [12]. This review provides the updated results on the benefits potentially associated with newly introduced antidiabetic drugs during infection.

## 2. Immunity Impairment in DM

Several pre-clinical and clinical studies have revealed a significant defect in both innate and adaptive immunity. Although some mechanisms are glycemia-independent [27], most of them rely on hyperglycemia and its metabolic effects, such as non-enzymatic glycation, generation of reactive oxygen species, and hyperactivity of the polyol pathway [28].

### 2.1. Neutrophils

Neutrophils are recognized as key elements to counteract infection, and DM impairs their recruitment as well as their killing capability. Several mechanisms also account for the dysfunction of adhesion, rolling, and chemotaxis [29,30,31]. CXCR2, a chemokine receptor expressed on neutrophils, was found downregulated during sepsis, thereby impairing neutrophil recruitment [27,32]. Moreover, since CXCR2 also controls the expression of the intracellular adhesion molecule-1 (ICAM) on endothelial cells (ECs), its decreased expression further weakens neutrophil recruitment at the inflammatory site [33]. Regarding phagocytosis, the main recognized abnormality is related to C3-mediated opsonization owing to complement glycation [34]. Both intracellular and extracellular killing mechanisms (involving the production of intracellular ROS [35,36], enzymatic degranulation [37], and inhibition of neutrophil extracellular traps (NET) formation [38]) are impaired in the hyperglycemic condition.

### 2.2. Macrophages

Chronic hyperglycemia also weakens macrophage adhesion and chemotaxis, antibacterial activity, and phagocytosis, damaging both the FCy receptor and the complement cascade [39,40]. Furthermore, the low-grade inflammation caused by hyperglycemia, insulin resistance, and obesity promotes macrophage differentiation towards their anti-inflammatory M2 phenotype, activating the IL-6 signaling pathway [41]. The homeostatic action of IL-6 in limiting inflammation represents a relevant impediment to the control of infections [42]. Figure 2 summarizes the most relevant DM-associated immune cell impairment.

### 2.3. Natural Killer Cells

Hyperglycemia-induced oxidative stress also compromises natural killer (NK) cell activity [43]. Importantly, a strong inverse linear relationship between their activity and HbA1c level has been reported [44,45].

### 2.4. Adaptive Immunity

The impact of DM on adaptive immunity is still debated since a few studies on T lymphocyte dysfunction have reported conflicting results. Preclinical studies have shown that the decreased expression of ICAM and E selectin in ECs impairs the recruitment of cytotoxic CD8+ T lymphocytes to sites of infection, resulting in more severe disease [46]. Dysregulation of the complement cascade has also been reported. As mentioned earlier, non-enzymatic glycation of C3 and C4 decreases opsonization [34]. In addition, the glycation of immunoglobulins can inhibit antigen recognition [47].

In conclusion, DM significantly impacts the immune system, resulting in a higher risk of infections. Several studies have shown defects in both innate and adaptive immunity. Hyperglycemia and insulin resistance are the most relevant factors contributing to the dysfunction of the immune system, which mainly involves neutrophils, macrophages, and natural killer cells.

## 3. Treatment-Associated Infections

As extensively discussed [8], a clear association between DM and increased infection-related mortality/morbidity is still uncertain. A recent review [6] investigating this specific topic has reported no difference in hospital mortality between diabetic and non-diabetic individuals, concluding that, rather than DM per se, DM-related co-morbidities and long-term complications might drive worse outcomes. Compared to healthy individuals, septic patients are connoted by the increased production of acute phase proteins, such as C reactive protein (CRP). No difference between serum levels of CRP in DM and non-DM patients has been documented [48].

The 2021 International Guidelines for the management of sepsis and septic shock [49], like the previous one (2016) [50], strongly recommended treating patients displaying blood sugar level (BGL) up to 180 mg/dL, thereby underlining the lack of studies on DM patients. Indeed, since hyperglycemia was not linked to increased ICU mortality, a different study proposed a Mean Blood Glucose (MBG) of between 140 and 190 mg/dL to avoid hypoglycemia and its adverse consequences during sepsis (such as increased oxidative stress, platelet aggregation, production of pro-inflammatory cytokines, and the expression of vascular adhesion molecules) [51,52]. In the following section, the relationship between single anti-diabetic agents and outcomes in DM patients with sepsis is discussed.

### 3.1. Metformin

Metformin, due to its pleiotropic effect [8] and impact on mitochondrial activity, autophagy, and immune modulation [53], appears to be a reliable and safe anti-diabetic drug during infections. A study by Gomez et al. [54] reported that metformin reduces the incidence of sepsis-induced AKI. Moreover, a lower incidence of mortality in patients treated with metformin prior to hospitalization for sepsis has also been shown [55,56,57]. Additionally, investigating the therapeutic impact of metformin against MRS, and multidrug-resistant (MDR) *Pseudomonas aeruginosa* in combination with antimicrobial agents, it was demonstrated that metformin synergizes with the majority of tested agents, with the highest antibiotic MIC reduction (93% in both cases) when combined with doxycycline and chloramphenicol [58]. The main concern related to metformin in septic patients relies on the risk of lactic acidosis, potentially worsening the already fragile clinical condition. However, a Meta-Analysis by Li et al. [57] analyzing 8195 patients, did not find a statistical difference in the level of serum creatinine and lactic acid between patients treated or not with metformin at pre-admission. Although validation is required, this observation opens a new scenario on metformin in hospitalized patients.

### 3.2. Insulin

Insulin still represents the first choice to lower BGL towards a safer value [59], however, its effectiveness could depend on specific settings. In fact, its anabolic effect might inhibit autophagy, thereby decreasing the antioxidant action [60], or contribute to antibiotic resistance by affecting biofilm growth. Patel et al. [61] showed that while insulin alone has no effect on the level of biofilm formation or cell growth, the presence of glucose significantly enhances both. This could be a trigger for the expression of biofilm formation and UTI, particularly in the presence of external catheters. Furthermore, this effect is increased by a temperature up to 37 °C, which is commonly experienced by septic patients. Using a preclinical diabetic model, Wei et al. [62] demonstrated that insulin promoted biofilm formation by activating the cyclic-di-GMP signaling pathway. This translated into delayed wound healing and increased antibiotic resistance against *P. aeruginosa* infection. Due to its effects on T cell proliferation and intermediary metabolism, several studies have proposed that insulin resistance could also impact susceptibility to H1N1 infection and the effectiveness of vaccination [63].

### 3.3. Glucagon-Like Peptide-1 Receptor Agonists (GLP-1) and Dipeptidyl Peptidase-4 Inhibitor (DPP4 Is)

Glucagon-like peptide-1 receptor agonists (GLP-1) and dipeptidyl Peptidase-4 Inhibitors (DPP4 is) have also been investigated in the last years, demonstrating a direct action on endotoxemia, independently of their glucose-lowering properties [64,65,66,67,68,69]. Specifically, it has shown an increase in biomarkers of inflammation, oxidative stress parameters, and endothelial dysfunction. It has also been shown that activation of GLP-1 receptors can promote B and T cell expansion, particularly toward Treg1 differentiation, thereby contributing to the impaired inflammatory response in patients with sepsis [51]. In addition, the massive activation of the endogenous GLP-1 system during sepsis has been proposed as a predictor of early death or persistent organ dysfunction [70,71], particularly in patients infected by Gram-negative bacteria [51,72].

### 3.4. SGLT2-Inhibitors (SGLT2i)

SGLT2-inhibitors (SGLT2i) are a class of antidiabetic drugs, which gained increasing interest for their proven long-term cardio and reno-protective effects [73,74,75]. The original concerns regarding a possible increase in UTIs are currently mitigated since it was limited to Dapaglifozin [76]. Moreover, compared to other active antidiabetic treatments, SGLT2i did not show a difference in the incidence of UTI [76,77,78]. Additionally, a recent systematic review and meta-analysis by Wang et al. [79] demonstrated that SGLT2i displays a powerful anti-inflammatory effect, recognized by the decrease in ferritin, leptin, and plasminogen activator inhibitor (PAI)-1. Although the mechanisms remain unclear, the anti-inflammatory action of Dapagliflozin and Empagliflozin was confirmed by efficacy tests and by the drop in morbidity and mortality observed in preclinical models of sepsis and renal injury [8,80,81,82]. Finally, a large meta-analysis (4568 citations, 26 trials with a total of 59,264 patients) by Li et al. [83] identified a significant reduction of the risk of pneumonia and septic shock in DM patients treated with SGLT2i. Certainly, future studies on SGLT2i in patients with septic shock are needed to better explore this promising antidiabetic class. Table 1 and Figure 3 summarize all these notions.

Since SARS-CoV-2 and tuberculosis frequently occur in DM patients, a detailed description of their link will be approached.

## 4. Diabetes and SARS-CoV-2 Infection

The severe acute respiratory syndrome coronavirus 2 (SARS-CoV-2), first identified at the beginning of 2020, has led to a global pandemic known as COVID-19. COVID-19 has affected more than 752 million people and caused more than 6.8 million deaths worldwide (updated to 28 January 2023) [84]. Since the outbreak of the pandemic, several medications, including vaccines, were administered to decrease transmission, hospitalization, and infection-associated death. Despite these efforts, a significant number of patients, particularly those with DM, hypertension, chronic obstructive pulmonary disease, obesity, and cardiovascular disease continue to experience fatal outcomes [85]. DM is recognized as a risk factor for poor outcomes, including progression to acute respiratory distress syndrome (ARDS) and mortality [86]. These patients are generally characterized by the presence of comorbidities such as retinopathy, kidney injury, poor metabolic control, or have a history of hospitalization for diabetic ketoacidosis or hypoglycemia in the past 5 years and are mostly treated with several anti-diabetic medications [87]. DM patients, particularly those with poor metabolic control, are at a higher risk of severe complications and death from COVID-19. SARS-CoV-2 infection also increases the risk of thromboembolism and is more likely to induce cardiorespiratory failure in DM patients than in non-DM individuals [88]. Additionally, the occurrence of DM onset after COVID-19 hospitalization has been reported [89,90].

### 4.1. Diabetes and Increased Susceptibility to COVID-19 Infection

Multiple mechanisms have been proposed to explain the increased complications and mortality observed in DM individuals. DM is associated with decreased phagocytic activity, neutrophil chemotaxis, T cell function, and lower innate and adaptive immune activities [91]. Hyperglycemia is an independent factor associated with a severe prognosis in individuals hospitalized for COVID-19 [92]. Hyperglycemia may rely on stress, inflammation, and disruption of beta cells, as well as steroid administration [93]. Hyperglycemia negatively impacts innate cell-mediated immunity [91] and perturbs the antiviral response by suppressing Th1/Th17 cell activation, inducing oxidative stress, and causing endothelial dysfunction [94]. Moreover, the presence of hyperglycemia at the time of hospital admission may result from the exacerbation of insulin resistance driven by the release of counter-regulatory hormones and cytokines, which in turn impact the immune response [95]. Glycation of proteins, microangiopathy of alveolar capillaries, and proteolysis of connective tissue, finally translate into the collapse of small airways during expiration [96]. Moreover, acute hyperglycemia increases the activity of the urinary angiotensin-converting enzyme 2 (ACE2), which in turn enhances the virulence of SARS-CoV-2 [97,98]. Indeed, SARS-CoV-2 binds to the ACE2 receptor, which plays a role in multiple molecular processes and regulates glucose levels [99]. ACE2 degrades angiotensin II and angiotensin I into the smaller peptides angiotensin-(1–7) and angiotensin-(1–9). Angiotensin-(1–7) has antioxidant and anti-inflammatory effects through the Mas receptor pathway, which can be altered in DM individuals [100]. In non-survivors, the pathway that regulates inflammation appears to be imbalanced, with a drop in angiotensin-(1–7) levels [101]. In conclusion, hyperglycemia activates inflammatory pathways and exacerbates oxidative stress, weakening the immune system [102]. DM patients with COVID-19 exhibit an imbalanced anti-inflammatory and pro-inflammatory T cell ratio, characterized by the over-activation of the Th1 and Th17 subsets [103]. This results in a high level of C-reactive protein (CRP), pro-calcitonin, ferritin, and IL-6, which contribute to the hyper-immune response denoted as a cytokine storm [104] (Figure 4).

### 4.2. Antidiabetic Agents and SARS-CoV-2

Prognostic benefits in DM patients can be accomplished by proper glycemic control, which also results from a balanced diet, physical activity, and consistent monitoring of blood glucose and blood pressure levels [102,105]. Increasing evidence indicates that glycemic control is crucial for COVID-19 hospitalized patients [86]. Among the antidiabetics, metformin, DPP4is and GLP-1Ras, SGLT2is, and insulin have been administered in patients with COVID-19. Administration of metformin has been shown to lower the incidence of mortality and hospitalization in COVID-19 DM patients [106]. Among the pleiotropic effect of metformin, its anti-inflammatory action is included [107]. Specifically, it improves ACE2 stability by hampering its ubiquitination and proteasome-mediated degradation [108] and leads to the reduced production of reactive oxygen species, oxidative stress, and DNA damage [102]. Researchers have suggested a therapeutic effect of DPP4is in SARS-CoV-2 infection [102]. Recent studies demonstrated that DPP4 inhibitors possess anti-inflammatory, immunomodulatory, and anti-fibrotic features [109]. However, other studies have reported that DPP4is significantly increases the risk of hospitalization and intensive care unit admission [110]. GLP-1Ras act on the ACE2 and Mas receptor pathways and may prevent SARS-CoV-2 infection and modulate inflammation and fibrosis [111] Studies showed a significant reduction in mortality and hospital admission in patients treated with GLP-1Ras or GLP-1RAs pre-admission [110,112,113]. However, the introduction of GLP-1RAs in critically ill patients is not fully recommended based on their potential side effects, the need for titration, and the therapeutic window [114]. The impact of SGLT2 inhibitors on COVID-19 has yet to be fully established, and the occurrence of diabetic ketoacidosis during gliflozin administration must be considered with caution [115]. Recent studies have shown a statistically significant decrease in hospitalization in patients treated with SGLT-2 inhibitors [110]. However, in acutely ill hospitalized patients with COVID-19 and DM, insulin is recommended [116], based on its anti-inflammatory action and the ability to suppress ACE2 expression [117]. Patients with severe COVID-19 and DM frequently require higher doses of insulin [102], thereby multi-injection insulin therapy is considered the most appropriate therapeutic option [118].

## 5. DM and Tuberculosis

The link between DM and Tuberculosis (TB) has been investigated by immunological and epidemiology studies. Chronic hyperglycemia represents a major risk factor for TB infection, disease severity, and treatment response by weakening the host immune response. TB still represents a major problem in low- and middle-income countries, particularly in the area of ongoing HIV epidemics, overcrowded living conditions, and inadequate healthcare systems [119]. At the same time, increasing industrialization, rapid urbanization, and aging populations are the major causes of the growing incidence of obesity and DM in emerging nations [3]. The association between DM and TB is becoming a global health problem and can be better understood using a syndemic model [120]. The interaction between DM and TB and the need for chronic care represents a significant burden for public healthcare systems. Poor glycemic control reflects inadequate access to effective diabetes care [121], which is associated with high rates of undiagnosed DM in low- and middle-income countries [122]. According to multiple observational studies, DM increases the risk of progression from latent TB to active disease threefold [123]. Therefore, socioeconomic disparities in DM diagnosis and lack of adequate patient care enhance TB susceptibility. Furthermore, DM not only increases the incidence of active TB but also increases the disease severity, with higher death rates and relapse after the accomplishment of antibiotic treatment [124]. Moreover, the presence of DM is associated with greater severity and delayed sputum conversion [125], which boost TB transmission. Finally, growing evidence indicates that DM is also associated with an increased risk of multi-drug resistant TB (MDR-TB) [126,127], further impairing the control of both epidemics.

### 5.1. Immune Mechanisms

A pathogenic model for TB susceptibility in DM patients has been proposed based on several studies [128]. Resident alveolar macrophages serve as the first line of immune defense against *Mycobacterium tuberculosis* bacilli (Mtb) in the lungs. Despite their inability to control Mtb replication, macrophages activate signals that recruit macrophages, dendritic cells, and neutrophils to present antigens to T-cells in lung-draining lymph nodes and prime the adaptive immune response. However, chronic hyperglycemia hampers leukocyte activity, particularly impairing the sentinel role of Mtb-infected macrophages [129]. Due to defective phagocytosis, reduced chemokine production, delayed recruitment of antigen-presenting cells (APC), and prolonged priming of adaptive immunity are commonly found. Therefore, Mtb replication occurs before T-cell activation. Once activated, the immune response promotes the release of pro-inflammatory cytokines and increases T cell proliferation [130,131,132], leading to increased disease severity, tissue damage, and poor outcomes (Figure 4). Overall, different mechanisms play a crucial role in increasing TB susceptibility, shortening survival, and increasing disease severity and recurrence rates in DM patients [133,134]. Future studies are needed to identify specific immune-metabolic pathways involved in the defective anti-tubercular response to be exploited as targeted approaches in DM patients.

### 5.2. Management of Tuberculosis in DM Patients

Combined diagnosis of DM and TB poses a significant challenge for the management of the disease. Studies on screening for active TB in DM patients have demonstrated weak results in terms of cost-effectiveness, with a major impact in high TB prevalence areas [135]. For this reason, a better risk stratification (including TB prevalence, history of TB, glycemic control, socioeconomic variables, and symptoms) is required for TB screening in DM patients. Additionally, screening for latent TB in DM patients, particularly in those with poor glycemic control, would identify a specific high-risk population who may benefit from prophylaxis. Unfortunately, the efficacy of preventive treatment in DM patients compared to non-diabetics remains undetermined, making it a priority and a future challenge. On the other hand, DM screening in TB patients is essential to control both infection and chronic complications associated with hyperglycemia. The availability of an adequate healthcare system and local conditions influence the choice of screening tests for DM in low- and middle-income countries [136,137]. Repeat fasting or random glucose tests are widely available, but results are commonly altered by transient hyperglycemia caused by active infection [138]. Point-of-cares for HbA1c measurement, as well as non-invasive advanced glycation end-product evaluation, could represent valid alternative approaches to screening DM, deserving a more accurate evaluation of the TB population [139]. The optimal treatment strategy for TB infection in DM patients is not yet established, and the standard antibiotic regimen is not tailored to comorbidities. As previously mentioned, DM patients have a higher rate of treatment failure, recurrence, and death. Some studies have shown lower serum concentrations of rifampicin [140] in diabetics and overweight patients, suggesting hidden pharmacokinetic variations of TB drugs. Increasing the dose of rifampicin or extending the time of treatment could improve TB outcomes [141,142], however, drug toxicity should be considered. Additionally, anti-diabetic drugs are influenced by TB antibiotic regimens. For example, rifampicin enhances the hepatic metabolism of all sulphonylureas, leading to complex dosing and increased adverse effects such as hypoglycemia. Nevertheless, sulphonylureas remain the most prescribed hypoglycemic drug in low- and middle-income countries due to their cost-effectiveness [143]. Metformin effects could be also impaired by rifampicin, owing to the increase in hepatic uptake and major glucose-lowering effects [144]. New oral antidiabetic drugs, such as GLP-1RAs, DPP4is, and SGLT2, have not shown clinically significant interactions with anti-TB drugs [145]. In contrast, even if insulin does not undergo hepatic metabolism, its availability, storage, delivery, and cost represent the main drawbacks to its administration in poor economic areas.

## 6. Conclusions and Future Perspective

In conclusion, DM is a world-spreading health problem and is associated with an increased risk of several infections. Chronic hyperglycemia impairs the function of several components of the immune system, thereby increasing the risk of infection-related morbidity and mortality. TB represents a long-lasting recognized infection associated with DM, while the SARS-CoV-2 pandemic has highlighted the risk of severe complications and death in DM patients. Attaining an early DM diagnosis and its related infections, managing patients to obtain proper glycemic control, and improving the selection of drugs lacking clinical interactions appear mandatory to reduce the burden of infection in DM patients. No definitive data are available on the safety of using newly introduced antidiabetic drugs in sepsis, thereby this topic should be a future challenge. Finally, the impact of hyperglycemia in impairing the immune system has spurred clinicians to search for and ascertain the beneficial effects of drugs administered to DM patients during sepsis in rescuing the immune defense. Data so far available have suggested that the prevention and reduction of infection severity may be considered the most relevant benefits. However, since randomized clinical trials are still an unmet need, tailored therapeutic strategies to manage infections and improve patient outcomes in DM still remain an open question.

## Figures and Tables

**Figure 1 biomedicines-11-00971-f001:**
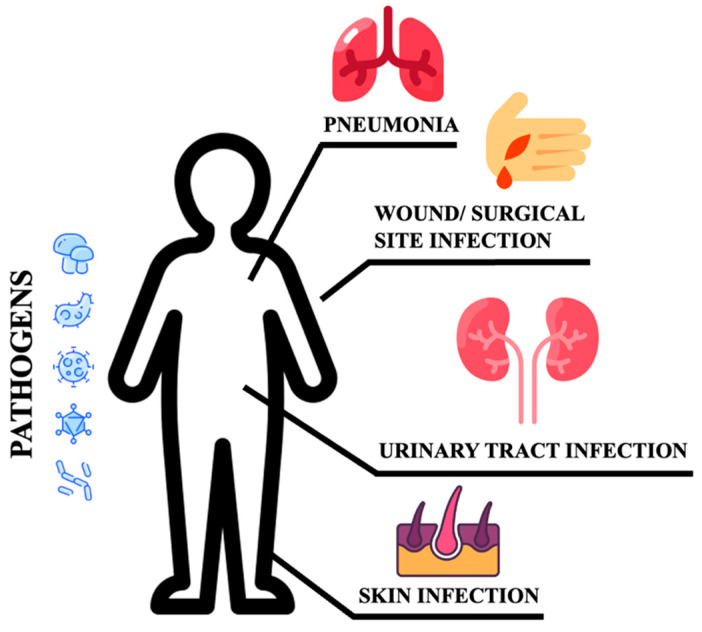
Sites of infections.

**Figure 2 biomedicines-11-00971-f002:**
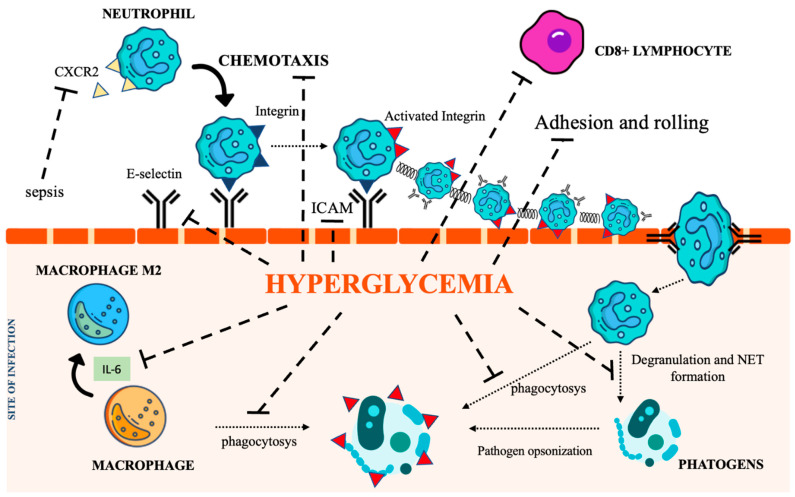
The immunological pathways under the control of hyperglycemia. CXCR2 (C-X-C Motif Chemokine Receptor 2), IL-6 (interleukin 6), NET (Neutrophil extracellular traps), ICAM (intercellular adhesion molecules). Icon design by icons8.

**Figure 3 biomedicines-11-00971-f003:**
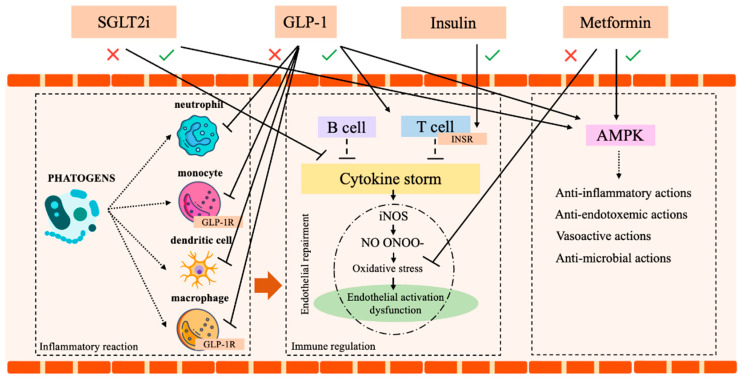
Immune system and insulin, metformin, SGLT2 and GLP-1 action. AMPK (AMP-activated protein kinase), SGLT2 (sodium-glucose co-transport 2 inhibitors), GLP-1 (Glucagon-like peptide-1 receptor agonists), GLP-1R (Glucagon-like peptide-1 receptor), T reg1 (T regulator 1 lymphocytes), INSR (Insulin receptor). Icon design by icons8.

**Figure 4 biomedicines-11-00971-f004:**
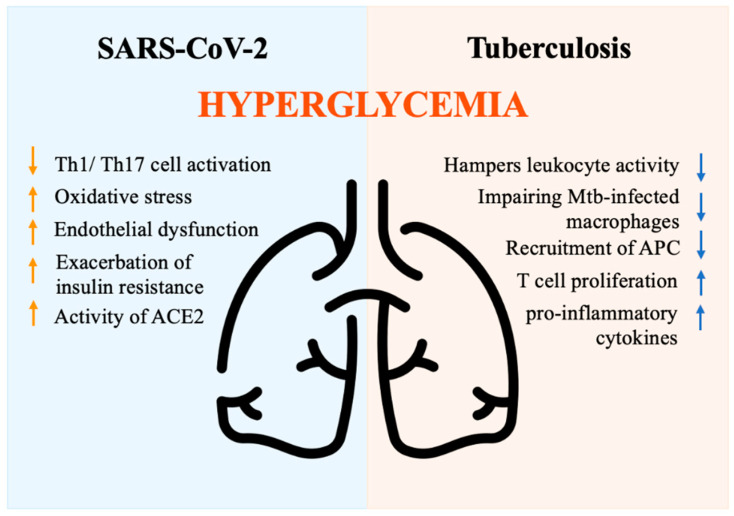
Immune response driven by hyperglycemia during SARS-CoV-2 and tuberculosis infections. ACE2 (Angiotensin-converting enzyme 2), APC (Antigen-presenting cells), Mtb (*Mycobacterium tuberculosis* bacilli), Th1/Th17 (Lymphocytes T helper 1 e 17), Icon design by icons8.

**Table 1 biomedicines-11-00971-t001:** Effects of anti-diabetic medications on immune modulation and inflammation.

Drugs	Author (Year)	Article Typology	Key Outcomes
Metformin	Costantini et al. (2021) [8]	Review Article	Metformin may exert important pleiotropic effects, involving the regulation of lactate metabolism and adenosine monophosphate-activated protein kinase (AMPK) activation, and produce anti-inflammatory, anti-endotoxemic, vasoactive and antimicrobial actions
	Bharath et al. (2021) [53]	Mini-Review	Metformin effect on mitochondrial function (inhibiting mitochondrial ROS and calcium-mediated activation of IL-6), autophagy, and immune modulation significantly impacts inflammation, independent of its role in blood glucose control.
	Gómez et al. * (2022) [54]	Retrospective cohort study	Exposure to metformin (*n* = 599) vs. not exposure (*n* = 2092) was associated with decreased 90-day mortality (71/599, 11.9% vs. 475/2092, 22.7%; OR, 0.46; 95% CI, 0.35–0.60), reduced severe acute kidney injury (50% vs. 57%; OR, 0.75; 95% CI, 0.62–0.90), lowered Major Adverse Kidney Events at 1 year (OR, 0.27; 95% CI, 0.22–0.68), and increased renal function recovery (95% vs. 86%; OR, 6.43; 95% CI, 3.42–12.1).
	Yen et al. (2022) [56]	Retrospective cohort study	In patients with DM, metformin displays no significant differences in the risks of UTI, recurrent UTI, or sepsis. However, it was associated with a lower risk of death due to UTI or sepsis than metformin non-user (*p* = 0.002).
	Li et al. (2021) [57]	Systematic Review and Meta-Analysis	At preadmission metformin use had lower mortality rate (OR, 0.74; 95% CIs, 0.62–0.88, *p* < 0.01) in patients with sepsis and DM. No statistically significant differences in the serum creatinine (*p* = 0.84) and lactic acid (*p* = 0.07) between preadmission metformin use and non-metformin use were reported.
	Masadeh et al. (2021) [58]	Pharmaceutical in vitro testing	In MRSA (ATCC 33,591) and MDR-*Pseudomonas aeruginosa* (ATCC BAA-2114) infection, combining metformin with the antibacterial agents had either synergetic or additive effects.
Insulin	Van Niekerk (2017) [60]	Viewpoint Article	An increase glucose levels might be adaptive in the short term (maintaining biosynthetic activities, supporting immune response during an infection), but may exert negative effects (mitochondrial and innate immune cell dysfunctions) in chronic settings. Insulin can inhibit autophagy that plays a pivotal role in both host defense and cell survival.
	Patel et al. (2021) [61]	Pharmaceutical in vitro testing	*E. coli* biofilm formation is insulin concentration dependent and is also influenced by oxygen concentration and temperature.
	Wei et al. (2019) [62]	Animal in vivo study/Pharmaceutical in vitro testing	Insulin did not promote the growth of *P. aeruginosa*. Insulin decreases the clearance of *P. aeruginosa* by inhibiting the Th1-type immune response and promoting biofilm formation by enhancing Th2-type polarization.
	Tsai et al. (2018) [63]	Review Article	Insulin receptor (INSR) on T cells supports cytokine production, effector cell differentiation, proliferation, nutrient uptake (and associated glycolytic and respiratory capacities), and boosts migration/recruitment to target organs. INSR deficiency compromises both CD4+ and CD8+ T cell compartments during influenza infection.
GLP-1 Ras/DPP-4i	Steven et al. (2015) [64]	Animal in vivo study	Linagliptin, liraglutide (and to a minor extent sitagliptin) therapy suppress LPS-induced inflammatory pathways (e.g., iNOS induction and activation, leukocyte activation, DC maturation and biomarkers of inflammation) in endotoxemic rats, through a GLP-1-mediated decrease of iNOS expression as well as the activation of AMPK as central survival pathway.
	Helmstädter et al. (2012) [65]	Animal in vivo study	Liraglutide displays both antioxidant and anti-inflammatory properties conferring vasoprotection in polymicrobial septic mice (lowering TNFα, IL-6, iNos and ICAM1 mRNA levels, attenuating elevated Nox2 protein)
	Steven et al. (2017) [66]	Animal in vivo study	GLP-1 receptor activation in platelets by linagliptin and liraglutide strongly attenuated endotoxemia-induced microvascular thrombosis and mortality by a cAMP/PKA-dependent mechanism, preventing systemic inflammation, vascular dysfunction, and end organ damage.
	Kröller-Schön et al. (2012) [67]	Animal in vivo study	Linagliptin, over all the other DDP-4i, demonstrated pleiotropic vasodilatory, antioxidant, and anti-inflammatory properties independent of its glucose-lowering properties. Linagliptin improved endothelial function by the reduction of leucocyte adhesion to endothelial cells in the presence of LPS.
	Al Zoubi et al. (2018) [68]	Animal in vivo study	Linagliptin significantly reduced sepsis-related cardiac, liver, kidney, and lung injury, by reducing NF-kB activation and iNOS expression in the heart, with lower serum inflammatory cytokine levels. Most notably, inhibition of NF-kB reduced organ dysfunction/injury associated with sepsis in mice with pre-existing T2DM.
	Wang et al. (2022) [69]	Animal in vivo study	Linagliptin exerted anti-inflammatory and anti-thrombotic effects independently of its effect on blood glucose level (inhibition of IL-1β and ICAM-1 expression, attenuation of tissue factor expression via the Akt/endothelial nitric oxide synthase phosphorylation)
	Yang et al. (2021) [51]	Review Article	GLP-1R is expressed in macrophages and monocytes and can inhibit the release of inflammatory factors. GLP-1R on can promote B- and T-cell proliferation, especially the expansion of Treg1, to inhibit systemic inflammatory response in sepsis patients. Thus, elevated endogenous GLP-1 levels are closely associated with worse outcomes. Therefore, since the GLP-1R is widely distributed in humans, GLP-1Ras have protective effects on multiple organs.
SGLT2is	Donnan et al. (2019) [76]	Systematic review and meta-analysis	When compared with placebo, SGLT2 inhibitors were found to be significantly protective against AKI (RR = 0.59; 95% CI 0.39 to 0.89), while no difference was found for ketoacidosis or UTI. Subgroup analysis showed an increased risk of UTI with dapagliflozin only (RR 1.21; 95% CI 1.02 to 1.43).
	Dave et al. (2019) [77]	Population-based cohort study	SGLT2is, when compared to DPP4i or a GLP-1RAs did not contribute to the severity of UTI events.
	Wiegley et al. (2022) [78]	Review Article	Despite SGLT2is-related glycosuria the increased urinary flow secondary to these medications’ osmotic effect has been proposed to explain the lack of clinically significant UTI. However, caution is required when SGLT2i agents are administered in patients with abnormal urinary flow (e.g., obstruction of urinary tract)
	Wang et al. (2022) [79]	Systematic review and meta-analysis	Compared to placebo or standard DM therapies, SGLT2is groups had reduced levels of ferritin (Standardized Mean Difference SMD −1.21; 95% CI: −1.91, −0.52, *p* < 0.001), C-reactive protein (SMD: 0.25; 95% CI: −0.47, −0.03, *p* = 0.02), leptin (SMD: −0.22; 95% CI:−0.43, −0.01, *p* = 0.04) and PAI-1 (SMD: −0.38; 95% CI: −0.61, −0.15, *p* = 0.001).
	Kıngır et al. (2019) [80]	Animal in vivo study	Dapagliflozin reduced oxidative stress (MDA), increased antioxidant levels (GSH), and reduced inflammation (MPO) in the kidney (*p* < 0.05). Dapagliflozin also decreased oxidative stress (MDA) in lung tissue and decreased inflammation (MPO) in lung and liver tissue (*p* < 0.05), although the effect was less relevant than in the kidney.
	Chi et al. (2021) [81]	Animal in vivo/in vitro study	Dapagliflozin attenuated endotoxin shock associated AKI and decreased the release of inflammatory cytokines in diabetic mice.
	Maayah et al. (2020) [82]	Animal in vivo study	Empagliflozin reduces mortality and inflammation in mice with established sepsis preventing renal injury, through the suppression of both local and systemic cytokine and chemokine release
	Li et al. (2022) [83]	Systematic review and meta-analysis	Compared with placebo, SGLT2is significantly reduced the risk of pneumonia (pooled RR 0.87, 95% CI 0.78–0.98) and septic shock (pooled RR 0.65, 95% CI 0.44–0.95).

* Montoya (2023) raised a few questions about the data-collection and the possible non-heterogeneity of the sampled patients.

## Data Availability

Not applicable.

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
