# Peer review of "Insight on Infections in Diabetic Setting"

_biomedicines, 2023, doi:10.3390/biomedicines11030971_

Round 1

Reviewer 1 Report

The article “Insights on Infections in Diabetic Setting” by Pari, B., et al describes the mechanisms accounting for the increased susceptibility of DM patients to infections with particular reference to the impact of hypoglycemic agents.  With various examples, the authors have pointed that the risk and mortality associated with infectious diseases are high in DM patients, implying that infections should be considered among the most common DM complications.  Furthermore, the authors have also shown that hyperglycemic state significantly impacts the immune system, resulting in a higher risk of infections. Several studies have shown defects in both innate and adaptive immunity due to diabetes mellitus. Hyperglycemia and insulin resistance are the most relevant factors contributing to the dysfunction of the immune system, which involves neutrophils, macrophages, and natural killer cells.

 In a separate section, the authors conclude that hyperglycemia activates inflammatory pathways and exacerbates oxidative stress. DM patients with infected with SARS-CoV-2 exhibit an imbalanced anti-inflammatory and proinflammatory T-cells ratio, characterized by the over activation of the Th1 and Th17 subsets. This imbalance results in a high level of C-reactive protein (CRP), pro-calcitonin, ferritin, and IL-6, which contribute to the hyper-immune response, also known as, cytokine storm.  Authors have also discussed about the DM and TB as a part of this review.

 Even though the authors have covered recent literature and provided information about the use of pharmacological agents and their impact on various outcomes, the review currently lacks the following

 1.     Appropriate figures explaining the concepts discussed in each section

2.     Future directions that can be explored

 In addition, the authors must describe how different this review is from several other reviews with similar theme. 

Minor comments

Page 8: SARS-COV-2 should be SARS-CoV-2

Author Response

We thank the Reviewer for His/Her appreciation and positive comments

Comments and Suggestions for Authors

The article “Insights on Infections in Diabetic Setting” by Pari, B., et al describes the mechanisms accounting for the increased susceptibility of DM patients to infections with particular reference to the impact of hypoglycemic agents.  With various examples, the authors have pointed that the risk and mortality associated with infectious diseases are high in DM patients, implying that infections should be considered among the most common DM complications.  Furthermore, the authors have also shown that hyperglycemic state significantly impacts the immune system, resulting in a higher risk of infections. Several studies have shown defects in both innate and adaptive immunity due to diabetes mellitus. Hyperglycemia and insulin resistance are the most relevant factors contributing to the dysfunction of the immune system, which involves neutrophils, macrophages, and natural killer cells.

 In a separate section, the authors conclude that hyperglycemia activates inflammatory pathways and exacerbates oxidative stress. DM patients with infected with SARS-CoV-2 exhibit an imbalanced anti-inflammatory and proinflammatory T-cells ratio, characterized by the over activation of the Th1 and Th17 subsets. This imbalance results in a high level of C-reactive protein (CRP), pro-calcitonin, ferritin, and IL-6, which contribute to the hyper-immune response, also known as, cytokine storm.  Authors have also discussed about the DM and TB as a part of this review.

Even though the authors have covered recent literature and provided information about the use of pharmacological agents and their impact on various outcomes, the review currently lacks the following

  1. Appropriate Figures explaining the concepts discussed in each section

As suggested by the Reviewer Figures have been included in the present version of the Ms.

  1. Future directions that can be explored

As kindly suggested by the Reviewer a sentence reporting perspective has been added in the present version of the Ms.

In addition, the authors must describe how different this review is from several other reviews with similar theme.

We thanks the Reviewer for the suggestion. This issue has been described in the Abstract section and at the end of the Introduction.

Minor comments

Page 8: SARS-COV-2 should be SARS-CoV-2

Many thanks, correction has been provided

Reviewer 2 Report

Insights on infections in diabetic setting

The article takes up a very important clinical issue, which is the relationship between diabetes and infections. The authors consider various aspects of this relationship, including the relationship between impaired immunity in glycemic milieu and the effects of antidiabetic agents on immune status.

Major comments:

The authors point out how the type of bacterial flora changes over time during hospitalization, but they do not compare this relationship with people without diabetes, nor do they explain why this fact.

In the introduction, there is no clear description of why the symptoms of infection in diabetes are diagnosed with a delay and what is its clinical significance, and there is no indication of the underestimated diagnostic significance of inflammatory markers.

The table presenting the results of studies evaluating the relationship of infection and immune status with the use of various antidiabetic molecules is extremely valuable. In the descriptive part devoted to drugs, there is no clear summary in which clinical situations, e.g. GLP-1 agonists should not be used and whether they can be used in inflammatory conditions, taking into account the description of their effect, e.g. on endotoxemia.

There is also no description of the characteristics of infections associated with the presence of diabetic wounds

Mino comments:

The authors do not group individual thematic sections, but describe individual issues one by one without clearly arranging them into subchapters. An example is chapter 6 and 7, which should be sub-chapters of chapter 5. Similarly, chapter 8 should be a separate chapter, and 9 and 10 should be sub-chapters.

The title of chapter 3 should specify prognosis – “of what/”, and chapter 4 treatment “of DM”-associated infections, although in fact this chapter deals with the impact of antidiabetic drugs on the course of infection.

After completing the above data, the article is worth publishing

Author Response

We thank the Reviewer for His/Her appreciation and positive comments

The article takes up a very important clinical issue, which is the relationship between diabetes and infections. The authors consider various aspects of this relationship, including the relationship between impaired immunity in glycemic milieu and the effects of antidiabetic agents on immune status.

Major comments:

The authors point out how the type of bacterial flora changes over time during hospitalization, but they do not compare this relationship with people without diabetes, nor do they explain why this fact.

We thank the Reviewer for the suggestion, all data compare DM patients and non DM patients.

In the introduction, there is no clear description of why the symptoms of infection in diabetes are diagnosed with a delay and what is its clinical significance, and there is no indication of the underestimated diagnostic significance of inflammatory markers.

As suggested by the Reviewer this issue has been included in the present version of the Ms.

The table presenting the results of studies evaluating the relationship of infection and immune status with the use of various antidiabetic molecules is extremely valuable. In the descriptive part devoted to drugs, there is no clear summary in which clinical situations, e.g. GLP-1 agonists should not be used and whether they can be used in inflammatory conditions, taking into account the description of their effect, e.g. on endotoxemia.

As suggested by the Reviewer this issue has been included in the present version of the Ms.

There is also no description of the characteristics of infections associated with the presence of diabetic wounds

As suggested by the Reviewer this issue has been included in the present version of the Ms.

Mino comments:

The authors do not group individual thematic sections, but describe individual issues one by one without clearly arranging them into subchapters. An example is chapter 6 and 7, which should be sub-chapters of chapter 5. Similarly, chapter 8 should be a separate chapter, and 9 and 10 should be sub-chapters.

As suggested by the Reviewer the Ms has been revised accordingly.

The title of chapter 3 should specify prognosis – “of what/”, and chapter 4 treatment “of DM”-associated infections, although in fact this chapter deals with the impact of antidiabetic drugs on the course of infection.

We thank the Reviewer for this comment. Prognosis has been removed.

Round 2

Reviewer 1 Report

Authors have adequately addressed the comments from my previous review report.